# Altered Membrane Expression and Function of CD11b Play a Role in the Immunosuppressive Effects of Morphine on Macrophages at the Nanomolar Level

**DOI:** 10.3390/ph16020282

**Published:** 2023-02-13

**Authors:** Peng-Cheng Yu, Cui-Yun Hao, Ying-Zhe Fan, Di Liu, Yi-Fan Qiao, Jia-Bao Yao, Chang-Zhu Li, Ye Yu

**Affiliations:** 1School of Basic Medicine and Clinical Pharmacy and State Key Laboratory of Natural Medicines, China Pharmaceutical University, Nanjing 210009, China; 2Interventional Cancer Institute of Chinese Integrative Medicine, Putuo Hospital, Shanghai University of Traditional Chinese Medicine, Shanghai 200062, China; 3State Key Laboratory of Utilization of Woody Oil Resource, Hunan Academy of Forestry, Changsha 410004, China

**Keywords:** morphine, opioid receptors, immunosuppression, macrophage phagocytosis, cd11b, thymopentin

## Abstract

Morphine, one of the most efficacious analgesics, is effective in severe pain, especially in patients with concomitant painful cancers. The clinical use of morphine may be accompanied by increased immunosuppression, susceptibility to infection and postoperative tumor metastatic recurrence, and the specific mechanisms and clinical strategies to alleviate this suppression remain to be investigated. Expression of CD11b is closely associated with the macrophage phagocytosis of xenobiotic particles, bacteria or tumor cells. Here, we find that morphine at 0.1–10 nM levels inhibited CD11b expression and function on macrophages via a μ-opioid receptor (MOR)-dependent mechanism, thereby reducing macrophage phagocytosis of tumor cells, a process that can be reversed by thymopentin (TP5), a commonly used immune-enhancing adjuvant in clinical practice. By knocking down or overexpressing MOR on macrophages and using naloxone, an antagonist of the MOR receptor, and LA1, a molecule that promotes macrophage CD11b activation, we suggest that morphine may regulate macrophage phagocytosis by inhibiting the surface expression and function of macrophage CD11b through the membrane expression and activation of MOR. The CD47/SIRPα axis, which is engaged in macrophage-tumor immune escape, was not significantly affected by morphine. Notably, TP5, when combined with morphine, reversed the inhibition of macrophage phagocytosis by morphine through mechanisms that promote membrane expression of CD11b and modulate its downstream signaling (e.g., NOS2, IFNG, IL1B and TNFA, as well as AGR1, PDGFB, IL6, STAT3, and MYC). Thus, altered membrane expression and function of CD11b may mediate the inhibition of macrophage phagocytosis by therapeutic doses of morphine, and the reversal of this process by TP5 may provide an effective palliative option for clinical immunosuppression by morphine.

## 1. Introduction

Opioids such as morphine are widely used worldwide for clinical postoperative analgesia or other causes of severe pain. Clinical studies have found that the use of opioids (e.g., morphine, fentanyl) causes, in addition to some common side effects such as tolerance, addiction, respiratory depression, nausea, and constipation, an immunosuppressive effect that increases the risk of infection [1,2]. It is known that opioid addicts are at increased risk of infection, and the intersection between HIV infection and intravenous drug abuse has been identified [3,4]. In vitro and in vivo experiments, as well as epidemiological and clinical studies on different patient populations, have also shown that opioids such as morphine and fentanyl impair the function of macrophages, natural killer (NK) cells, and T cells, and weaken the intestinal barrier, inducing immunosuppression of the innate and adaptive immune systems, as well as damaging the mucosal barrier [5,6,7]. For example, morphine treatment initiated a downregulation of major histocompatibility complex II (MHC-II) expression on B cells as well as diminished antigen-presenting function, which inhibited T cell proliferation and promoted Th2 cell differentiation [8]. It has also been claimed that morphine inhibits macrophage chemotaxis and ROS production and inhibits TNF-α, IL-10 and NO production by LPS-stimulated macrophages. In vitro studies have found that morphine directly inhibits macrophage phagocytosis and reduces microbial killing activity [9,10]. Morphine has a high affinity for μ-opioid receptor (MOR) and a low affinity for other receptors [11]. One study found by using MOR gene-deficient mice that macrophage phagocytosis was not inhibited by morphine in MOR knockout (KO) mice compared to WT mice, suggesting that the specific receptor involved in immunosuppression is the MOR [12,13]. A dose–effect curve study using different selective receptor agonists and antagonists found that MOR agonists inhibit phagocytosis in a dose-dependent manner [14,15]. Despite many previous studies, the mechanism of how morphine regulates macrophage function through MOR still needs further investigation.

CD11b is a type 1 transmembrane glycoprotein of approximately 170 kDa in size that is highly expressed on myeloid cells (including tumor-associated macrophages, TAM) and regulates a broad range of immune responses, such as pathogen recognition, phagocytosis, and cell survival [16]. CD11b function is dependent on a cascade of inside-out and outside-in activation signaling processes, particularly tyrosine kinase activation [17]. CD11b is involved in adhesion contacts between many cells, such as monocytes, macrophages, NK cells and granulocytes [18] and has also been shown to mediate macrophage adhesion, migration, chemotaxis and accumulation during inflammation [19,20]. Interestingly, activation of CD11b negatively regulates the immune response of B lymphocytes, which suppress self-reactive B cells in systemic lupus erythematosus (SLE) [21]. CD11b also regulates pro- and anti-inflammatory signaling in macrophages and is therefore an ideal receptor for targeting TAMs as a means of controlling tumor growth. It has been shown that CD11b regulates macrophage polarization through pro-inflammatory macrophage transcription, thereby inhibiting immunosuppressive macrophage polarization and improving anti-tumor immune responses in mouse and human tumor models, which in turn improves survival [22,23,24]. However, it is also not very clear whether CD11b function is regulated by opioid receptors on macrophages during the activation of morphine, leading to weakened immunity.

Here, we propose a possible mechanism of morphine-induced immunosuppression, namely that morphine reduces the expression and function of CD11b on macrophages via MOR, which inhibits the phagocytosis of macrophages and thus affects tumorigenesis, progression, and prognosis [25,26]. This aspect of the study may provide new insights into the use of opioids such as morphine and the prognosis and treatment of tumor symptoms. Meanwhile, thymopentin (TP5) is considered as a potential drug for the treatment of immunodeficiency, tumors, and infections, significantly improving the cellular and humoral immune functions of the body [27,28,29]. As an immune system booster, TP5 has been used in patients with malignancies after radiation and chemotherapy, as well as in the elderly and immunodeficient patients to reconstitute immune function [30]. Here, we also demonstrate that TP5 could reverse the reduction of CD11b membrane expression and function on macrophages by morphine through MOR, thereby restoring the phagocytic function of macrophages. Here, therefore, we not only reveal a new mechanism of morphine inhibition of macrophage CD11b but also propose a solution for clinical remission.

## 2. Results

### 2.1. Nanomolar (nM) Morphine Directly Inhibits Macrophage Phagocytosis of Cancer Cells via MOR but Not by Altering the CD47/SIRPα Axis

Opioid receptor agonists or antagonists affect tumor growth by modulating macrophage function [31,32]. To further verify at the cellular level whether morphine reduces macrophage phagocytosis of tumors at therapeutic doses, we induced THP-1 cells into phagocytic macrophages with phorbol-12-myristate-13-acetate (PMA), followed by A549 cells labeled with the fluorescent dye probe Protonex^TM^ Red 600, and co-cultured with macrophages for 24 h. The results were obtained by fluorescence intensity recorded by fluorescence microscopy photographs, and morphine at 10 nM significantly attenuated phagocytosis of A549 by macrophages compared to the controls (13.1 ± 3.2% vs. 34.7 ± 4.6%, 10 nM morphine vs. Ctrl, respectively, *p* < 0.01, n = 3, unpaired *t*-test, Figure 1A,B). Additionally, the phagocytosis of A549 by macrophages was examined by flow cytometry in the concentration range of 0.001 nM~10 μM, and it was found that morphine concentration dependently inhibited the phagocytic effect of macrophages (EC_50_ (concentration producing half the efficacy) = 2.13 ± 0.16 nM, Figure 1C), which is the concentration required for morphine to activate MOR [33], indicating that direct inhibition of tumor phagocytosis by macrophages at nM levels of morphine is a possible phenomenon during clinical administration.

It has been shown that CD47, which is highly expressed on cancer cells, and regulatory protein α (SIRPα), a macrophage expressing receptor, form checkpoints for innate immunity, inhibiting macrophage-mediated phagocytosis and leading to immune escape of tumor cells [34]. We conjectured first whether morphine inhibits macrophage phagocytosis by altering the CD47/SIRPα axis. Therefore, we labeled tumor cells A549 and macrophages with APC-CD47 and PE-SIRPα flow antibodies, respectively, and their expression levels on the membrane were detected by flow cytometry (Figure 1D–G). 10 nM Morphine did not alter the expression of CD47 in A549 cells and SIRPα protein in macrophages (77.9 ± 0.9% and 65.9 ± 9.9%, for 0 and 10 nM morphine in the surface expressions of CD47, respectively, *p* > 0.05, unpaired *t*-test, n = 3, Figure 1D,E; 89.4 ± 0.6% and 87.7 ± 0.8% for 0 and 10 nM morphine in the surface expressions of SIRPα, respectively, *p* > 0.05, n = 3, unpaired *t*-test, Figure 1F,G). Moreover, treatment with 0.01 nM~10 μM morphine for 48 h did not change the proliferation viability of macrophages and A549 cells compared with the control group (*p* > 0.05, n = 3, one-way ANOVA followed by Dunnett’s multiple comparisons test, Figure 1H,I). These results suggest that direct inhibition of cancer cells phagocytosis by macrophages at nM levels is not attained through direct proliferation-inhibiting effects on cancers, nor through the modulation of the macrophage-tumor immune escape pathway, at least not primarily through the mechanisms described above.

### 2.2. Morphine Inhibits the Phagocytic Effect of Cancer Cells by Decreasing the Membrane Expression of CD11b on Macrophages

CD11b mediates multiple monocyte/macrophage responses in the immune inflammatory response, and it is tightly associated with phagocytosis of macrophages [35]. Therefore, we speculated that morphine may affect the phagocytosis of macrophages by affecting their CD11b functions. A significant decrease in CD11b mRNA expression levels was detected by qRT-PCR in macrophages induced with morphine in the concentration range of 0.1 nM~1 μM for 48 h (*p* < 0.001, n = 4, one-way ANOVA followed by Dunnett’s multiple comparisons test, Figure 2A).

To further verify the effect of morphine on macrophage CD11b membrane expression, we labeled macrophages with APC-CD11b antibody. Morphine significantly decreased the expression level of CD11b protein on macrophage membranes in the concentration range of 0.1~100 nM (46.0 ± 3.4%, 44.2 ± 6.3%, 36.2 ± 8.0%, 24.1 ± 3.3% and 23.0 ± 2.7%, for 0, 0.1, 1, 10, and 100 nM Morphine, respectively, *p* < 0.01, n = 3, one-way ANOVA followed by Dunnett’s multiple comparisons test; Figure 2C,D). Therefore, morphine inhibited the phagocytosis of A549 cells by macrophages, probably because it suppressed the expression of macrophage CD11b. Furthermore, the inhibition efficiency (EC_50_ = 2.13 ± 0.16 nM) of phagocytosis was comparable to the inhibitory potency of morphine on CD11b expression (EC_50_ = 1.44 ± 0.39 nM, Figure 2E).

### 2.3. Leukadherin-1 (LA1) Promotes Macrophage CD11b Activation to Alleviate the Inhibitory Effect of Morphine on Macrophage Phagocytosis

LA1 is a small molecule agonist of CD11b that enhances the innate immune response, and in vivo targeting of CD11b activation repolarizes TAMs and inhibits tumor growth by enhancing the pro-inflammatory immune response to tumors [23,36]. To verify whether LA1 could affect the inhibition of morphine on macrophage phagocytosis by regulating CD11b expression or function, we first examined the expression of genes related to the downstream signaling of macrophage CD11b activation and found that morphine inhibited the expression of macrophage immune activation genes, such as NOS2, IFNG, and IL1B, and TNFA mRNA expression was decreased (*p* < 0.05, n = 3, one-way ANOVA followed by Dunnett’s multiple comparisons test, Figure 3B). The expression of immunosuppressive genes such as AGR1, PDGFB, IL6, STAT3 and MYC was also elevated after morphine administration (*p* < 0.05, n = 3, one-way ANOVA followed by Dunnett’s multiple comparisons test, Figure 3A). Interestingly, the combination of LA1 and morphine reversed and restored the downstream signaling of macrophage CD11b activation compared to the morphine alone group (*p* < 0.05, n = 3, one-way ANOVA followed by Dunnett’s multiple comparisons test, Figure 3A,B).

Next, the expression of CD11b on the membrane of LA1 and morphine co-administered macrophages was examined 48 h later. Compared with the control group, administration of 4 μM and 8 μM LA1 alone did not change the proportion of CD11b positive cells (41.1 ± 2.0% (4 μM LA1), 41.3 ± 1.9% (8 μM LA1) vs. 38.8 ± 2.4% (Ctrl), respectively, *p* > 0.05, n = 3, one-way ANOVA followed by Dunnett’s multiple comparisons test, Figure 3C,D), whereas the proportion of CD11b+ positive cells was somewhat restored by the combined administration of LA1 and morphine (28.7± 3.8% (4 μM LA1 + 10 nM morphine), 27.0 ± 1.7% (8 μM LA1 + 10 nM morphine) vs. 20.1 ± 1.4% (10 nM morphine), respectively, *p* < 0.05, n = 3, one-way ANOVA followed by Dunnett’s multiple comparisons test, Figure 3C,D).

To further determine whether LA1 affects the phagocytosis of tumor cells by macrophages, we labeled cancer cells A549 with the fluorescent dye probe Protonex^TM^ Red 600 and co-cultured them with macrophages for 24 h. Macrophages pretreated with LA1 significantly restored phagocytosis of A549 cells compared with the morphine administration group alone (31.2 ± 4.1% (4 μM LA1 + 10 nM morphine) and 32.7 ± 4.7% (8 μM LA1 + 10 nM morphine) vs. 14.4 ± 3.7% (10 nM morphine), respectively, *p* < 0.01, n = 3, one-way ANOVA followed by Dunnett’s multiple comparisons test, Figure 3E,F). Meanwhile, the proportion of macrophages that phagocytosed A549 with the combination of LA1 and morphine were detected by flow cytometry and we found that inhibition could also be restored (24.9 ± 1.3% (4 μM LA1 + 10 nM morphine), 27.3 ± 1.7% (8 μM LA1 + 10 nM morphine), vs. 15.2 ± 2.0% (10 nM morphine), respectively, *p* < 0.01, n = 3, one-way ANOVA followed by Dunnett’s multiple comparisons test, Figure 3G,H).

These results confirm that LA1 promotes macrophage CD11b activation and reverses morphine-mediated inhibition of macrophage phagocytosis.

### 2.4. Nanomolar Morphine Inhibits Phagocytosis of Cancer Cells by Decreasing the Surface Expression of MOR and CD11b on Macrophage Membranes

Morphine at sub-nM to nM levels alone modulates macrophage CD11b function and reduces phagocytosis, while MOR is also expressed in macrophages, suggesting that morphine-mediated immunomodulation may be mediated through direct interaction with MOR on immune cells [37]. First, 48 h action of morphine on induced macrophages with a concentration range of 0.1 nM~1 μM morphine revealed a significant decrease in mRNA accompanying the CD11b gene and a significant decrease in the OPRM (MOR) gene (*p* < 0.001, n = 4, one-way ANOVA followed by Dunnett’s multiple comparisons test, Figure 2B).

Then, morphine at concentrations of 0.01 nM~10 μM was applied to PMA-induced macrophages labeled with Anti-MOR antibody and FITC-anti-mouse IgG1 antibody, and flow cytometric results showed that morphine also significantly reduced the expression level of MOR on macrophage membranes (EC_50_ = 5.53 ± 3.70 nM, Figure 4A). We then synthesized small interfering RNA (siRNA for MOR) and detected the transient knockdown efficiency by transient transfection of induced macrophages by qRT-PCR. The knockdown efficiency of the MOR siRNA group was above 85% compared with the control siRNA (*p* < 0.001, n = 3, unpaired *t*-test, Figure 4B, left). Knockdown of the OPRM gene significantly reduced the expression level of CD11b (*p* < 0.001, n = 3, unpaired *t*-test, Figure 4B, right).

Next, the surface expression of CD11b on macrophages after transient knockdown of OPRM gene was examined and the results showed that knockdown of the OPRM gene could significantly inhibit CD11b expression on the macrophage membrane compared with the control group (25.5 ± 4.6% vs. 42.2 ± 0.8%, MOR siRNA vs. Ctrl siRNA, respectively, *p* < 0.01, n = 3, unpaired *t*-test, Figure 4C,D).

Moreover, by overexpressing hMOR-WT after macrophages for 48 h, OPRM mRNA expression (*p* < 0.001, n = 4, unpaired *t*-test, Figure 4E) was found to be significantly elevated along with CD11b gene mRNA expression (*p* < 0.05, n = 3, unpaired *t*-test, Figure 4E). Moreover, further flow cytometric analysis showed that the overexpression of MOR also significantly increased the expression level of membrane protein of CD11b on macrophages (62.0 ± 3.4% vs. 39.5 ± 2.7%, hMOR-OE vs. NC, respectively, *p* < 0.001, n = 3, one-way ANOVA followed by Dunnett’s multiple comparisons test, Figure 4F,G). Similarly, the proportion of CD11b+ positive cells was significantly restored by combined morphine and MOR overexpression (55.1 ± 3.0% (hMOR-OE + 10 nM morphine) vs. 20.8 ± 2.9% (10 nM morphine), respectively, *p* < 0.001, n = 3, one-way ANOVA followed by Dunnett’s multiple comparisons test, Figure 4F,G).

These results suggest that morphine may contribute to its inhibition of macrophage phagocytosis by reducing the co-distribution of MOR and CD11b across the macrophage membrane.

### 2.5. The MOR Inhibitor Naloxone Reverses the Inhibitory Effect of Morphine on Macrophage CD11b and Rescues the Phagocytic Function of Macrophages

To further verify the relationship between opioid receptor activation and morphine inhibition of macrophage CD11b membrane expression and thus the restoration of its phagocytosis, we have chosen the opioid receptor antagonist, naloxone (NLX), for validation [38]. After pretreatment of macrophages with 10 μM NLX for 30 min followed by morphine at a final concentration of 10 nM for 48 h, NLX significantly increased membrane CD11b expression (34.4 ± 3.5%) compared with morphine at 10 nM alone (19.8 ± 3.8%) (*p* < 0.05, n = 4, *t*-test, Figure 5A,B).

Similarly, we tested whether NLX could also restore the morphine-induced decrease in phagocytic activity of macrophages at nM levels. The results of co-culture with macrophages and cancer cells A549 showed a significant increase in fluorescence intensity and phagocytosis ratio of macrophages towards cancer cell phagocytosis in response to NLX at 10 μM (38.9 ± 2.9%), and 20 μM (41.7 ± 1.1%) compared to morphine administration alone (16.6% ± 4.3%) and control (42.1 ± 2.3%) (*p* < 0.001, NLX + morphine vs. morphine alone, n = 3, one-way ANOVA followed by Dunnett’s multiple comparisons. test, Figure 5C,D). Meanwhile, we examined the proportion of macrophages that phagocytosed A549 with the combination of NLX and morphine and found that inhibition could also be restored (28.2 ± 2.8% (10 μM NLX + 10 nM morphine), 29.6 ± 2.6% (20 μM NLX + 10 nM morphine), vs. 15.5 ± 1.5% (10 nM morphine), respectively, *p* < 0.01, n = 4, one-way ANOVA followed by Dunnett’s multiple comparisons test, Figure 5E,F), suggesting that NLX almost completely reversed the immunosuppression of macrophages by morphine.

At the same time, co-administration of NLX and morphine suppressed the expression of CD11b immunosuppression-related genes AGR1, PDGFB, IL6, STAT3, and MYC (*p* < 0.05, n = 3, one-way ANOVA followed by Dunnett’s multiple comparisons test, Figure 5G), and significantly increased the expression levels of NOS2, IFNG, IL1B, and TNFA, which stimulate immune activation (*p* < 0.05, n = 3, one-way ANOVA followed by Dunnett’s multiple comparisons test, Figure 5H).

These results suggest that the opioid receptor inhibitor naloxone can reverse the morphine-induced decrease in macrophage CD11b expression and restore the phagocytic effect of macrophages. Thus, morphine could reduce macrophage CD11b surface expression by activating MOR, which is another explanation for its inhibitory effect on macrophage phagocytosis. It is also possible that the decrease in membrane expression of MOR (endocytosis) is due to the sustained activation of MOR [39].

### 2.6. Thymopentin (TP5) Reverses the Inhibitory Effect of Morphine on Macrophage Phagocytosis by Promoting the Surface Expression and Function of CD11b

It is also not well understood how morphine-mediated immunosuppression can be restored by combining it with other clinically used drugs. TP5 has very strong immunomodulatory activity and can significantly improve the cellular and humoral immune functions of the body, thus making it a potential drug for the treatment of primary or secondary immunodeficiency, tumors, and severe infections [28,30]. At the same time, TP5, as an immune booster for tumor treatment, can restore the immune system of tumor patients, and in combination with chemotherapy, it can likewise prevent immunosuppression caused by chemotherapy drugs.

Indeed, TP5 administration alone significantly increased the mRNA expression of OPRM at concentrations of 50, 100 and 200 μM (*p* < 0.05, n = 3, one-way ANOVA followed by Dunnett’s multiple comparisons test, Figure 6A) and CD11b (*p* < 0.05, n = 4, one-way ANOVA followed by Dunnett’s multiple comparisons test, Figure 6B).

Next, TP5 significantly increased the expression level of CD11b protein on macrophage membranes at concentrations of 50 (42.9 ± 1.5%), 100 (49.2 ± 3.0%) and 200 μM (53.0 ± 5.3%) compared to control (41.6 ± 2.5%) (*p* < 0.05, n = 3, one-way ANOVA followed by Dunnett’s multiple comparisons test, Figure 6C,D). Meanwhile, we examined the proportion of macrophages that phagocytosed A549 with TP5 administration alone and found that phagocytosis intensity was increased (28.0 ± 1.8% (100 μM TP5), 31.6 ± 1.3% (200 μM TP5) vs. 23.7 ± 2.8% (Ctrl), respectively, *p* < 0.01, n = 3, one-way ANOVA followed by Dunnett’s multiple comparisons test, Figure 6E,F).

Moreover, compared with the morphine alone group (25.2 ± 1.9%), the coadministration of 50 μM, 100 μM or 200 μM TP5 and 10 nM morphine for 48 h significantly increased the expression of CD11b on macrophage membranes at 100 (32.4 ± 3.3%), and 200 μM (36.0 ± 3.9%) (*p* < 0.05, n = 3, one-way ANOVA followed by Dunnett’s multiple comparisons test, Figure 6G,H), although 50 μM TP5 (27.0 ± 2.3%) did not alter the expression of CD11b-positive cells on macrophage membranes (*p* > 0.05, n = 3, one-way ANOVA followed by Dunnett’s multiple comparisons test, Figure 6G,H).

Similarly, qRT-PCR results showed that the combination of TP5 and morphine also increased the expression levels of two genes, CD11b and OPRM (Figure 6I,J). Pre-administration of TP5 also decreased the expression of CD11b immunosuppression-related genes AGR1, PDGFB, IL6, STAT3, and MYC (*p* < 0.05, n = 3, one-way ANOVA followed by Dunnett’s multiple comparisons test, Figure 6K) and significantly increased the expression levels of NOS2, IFNG, IL1B, and TNFA, genes that stimulate immune activation (*p* < 0.05, n = 3, one-way ANOVA followed by Dunnett’s multiple comparisons test, Figure 6L).

Further macrophage phagocytosis assays showed that although TP5 at 50 μM concentrations were not significant, the combination of 100 μM or 200 μM and 10 nM morphine revealed a concentration-dependent enhancement of macrophage phagocytosis by TP5 (22.6 ± 3.8% (100 μM TP5 + 10 nM morphine), 38.3 ± 8.0% (200 μM TP5 + 10 nM morphine), vs. 9.6 ± 1.4% (10 nM morphine alone), respectively, *p* < 0.01, n = 3, one-way ANOVA followed by Dunnett’s multiple comparisons test, Figure 7A,B). Notably, compared with the morphine administration group alone, TP5 at 200 μM almost completely restored macrophage phagocytosis intensity to the pre-morphine administration level (38.3 ± 8.0% vs. 39.4 ± 5.7%, 200 μM TP5 + 10 nM morphine vs. Ctrl, respectively, *p* > 0.05, n = 3, one-way ANOVA followed by Dunnett’s multiple comparisons test, Figure 7A,B). Meanwhile, the phagocytosis of A549 by macrophages was examined by flow cytometry, and it was found that TP5 could completely reverse the inhibition of function on macrophages by morphine (*p* > 0.05, n = 4, one-way ANOVA followed by Dunnett’s multiple comparisons test, Figure 7C,D).

Our previous study showed that TP5 can inhibit the stemness of colorectal cancer stem cells via nicotinic acetylcholine receptors (nAchRs) [40], and the TP5-mediated changes in macrophage function were further verified here also by the acetylcholine receptor antagonist D-Tubocurarine chloride pentahydrate (TUB). Pretreatment with 150 μg/mL (24.0 ± 1.8%) or 200 μg/mL (22.1 ± 2.9%) of TUB significantly attenuated the reversal of morphine-induced CD11b surface expression decrease by TP5′s treatments (*p* < 0.01, vs. 200 μM TP5 + 10 nM morphine (34.0 ± 1.1%), n = 3, one-way ANOVA followed by Dunnett’s multiple comparisons test, Figure 7E,F).

Together, these results suggest that TP5 reversed the inhibitory effect of morphine on macrophage phagocytosis, mainly by promoting membrane expression of CD11b and regulation of downstream signaling, a process in which nAchRs are also partially involved.

## 3. Discussion

Here, we find that morphine inhibited the phagocytic effect of macrophages on tumor cells in a system of macrophage and tumor cell co-culture, and that this inhibition was somewhat morphine concentration dependent (at nanomolar level). This inhibition of phagocytosis was not achieved by inhibiting the proliferative activity of macrophages as well as cancer cells A549, or the innate immunity CD47/SIRPα axis. Our results suggest that the inhibition of phagocytosis by morphine is due to its suppression of macrophage CD11b expression and function, and that the reduced phagocytic effect is consistent with a potency to suppress CD11b expression. In addition, LA1 can promote CD11b activation downstream to stimulate the expression of immune activation-related genes and inhibit immune suppression-related genes, thus reversing the immunosuppressive effects of morphine. Further studies with knockdown, antagonist inhibitors, and overexpression of MOR on macrophages suggest that morphine may inhibit phagocytosis of macrophages by reducing macrophage CD11b expression through MOR’s surface expression and activation. We found that in combination with morphine, TPF, which is an immune-enhancing adjuvant, reversed the inhibitory effect of morphine on macrophage phagocytosis and restored the macrophage phagocytic effect, mainly by promoting CD11b expression and activation of downstream signaling. Thus, CD11b and MOR jointly contribute to the immunosuppressive effects of morphine, with an association between them, and TP5 can regulate both, thereby reversing the immunosuppression of morphine.

We suggest a possible mechanism for morphine-induced immunosuppression and decreased function, namely that morphine inhibits CD11b expression and function on macrophages via MOR activation, thereby suppressing macrophage phagocytosis. However, we do not know whether morphine only affects changes in CD11b expression during monocyte-to-macrophage conversion or also directly affects CD11b as a membrane receptor-mediated downstream immune-related signaling, which our current findings cannot distinguish. Additionally, whether there is a strong association between decreased membrane expression of MOR and decreased membrane expression of CD11b, and whether there is a direct interaction between the two, remains to be further investigated. The decrease in macrophage phagocytosis induced by nM levels of morphine may be due to two mechanisms: decreased expression of MOR and CD11b in the membrane, and activation of MOR receptors; it remains to be confirmed whether these two mechanisms interact or are causal. It is also not very clear whether the decreased membrane expression of MOR is caused by the sustained activation of MOR, which leads to a decreased surface expression of CD11b, or whether other factors mediate this process. Considering that the effect of LA1 activation does not fully restore morphine-suppressed CD11b surface expression, we speculate that there may be other mechanisms mediating this effect, which also need further investigation. This mode of action might not depend primarily on G-protein downstream signaling from the MOR receptors and could serve to modulate the immunosuppressive effects of morphine without affecting its analgesic effects.

The relationship between prolonged opioid intake and tumor development in non-cancer chronic pain medication is not well understood, and there is growing evidence that patients in these settings are at risk for cancer [15]. The common dose of clinical postoperative morphine administered subcutaneously is 5–10 mg/dose and 10–40 mg/day in adults (up to 60 mg or more depending on the severity of cancer pain), and the blood concentration of morphine in vivo is ~50–100 nM [41]; when administered intravenously, the common dose in adults for analgesia is 1–10 mg. Studies have shown that an intravenous infusion of morphine 10 mg has an initial blood concentration of 100–200 ng/mL, which converts to approximately 350–700 nM; the lowest effective analgesic blood concentration of morphine in vivo is 10–50 ng/mL, which converts to approximately 35–175 nM; blood concentrations in the range 1.8–38 nM were detected 1 h after oral administration of 10 mg morphine, with a mean blood concentration of 5.8 nM [42,43], peak serum concentrations of morphine reached 70 to 80 ng/mL within 10 to 20 min after intramuscular administration of 10 mg [44]. These concentrations of clinical use of morphine are within the range of the present study, which validates our hypothesis of clinical immunosuppression caused by morphine, of which immunosuppression of macrophages is one of the causes. In clinical practice, TP5 might be considered in combination with morphine, which can effectively circumvent the risk of reduced immune function due to long-term morphine use in oncology patients.

Another point is that immune cells have been shown to secrete acetylcholine, dopamine, and gamma-aminobutyric acid (GABA), which mediate immune signaling and play important immunomodulatory roles [45,46,47,48]. In this study, pretreatment with TUB significantly attenuated the reversal of morphine-induced CD11b surface expression decrease by TP5’s treatments (Figure 7), suggesting a process in which nAchRs are also partially involved.

Another interesting question is whether the current results have the potential to be applied clinically to retain the analgesic effect of morphine while reducing its effect on immunosuppression. We have confirmed that there are no reports of TP5 affecting the clinical application of morphine analgesia. Only one study in an animal model found that TP5 treatment resulted in a slight decrease in sensitivity to painful stimuli in rats. TP5 enabled tumor-bearing animals to recover diminished behavioral activity and increased resistance to stressful stimuli and pain [49], suggesting that TP5 can exhibit a slight analgesic effect. Furthermore, the advantage of TP5 over naloxone is that it may only improve the immunosuppressive effects of morphine without affecting the analgesic effects of morphine.

Finally, for the clinical use of TP5, clinical administration is mainly intramuscular or subcutaneous and can be 50 mg per day [50]. A study evaluated the clinical efficacy and tolerability of high-dose intravenous TP5 in 16 patients with melanoma [51]. Patients received 1 g of TP5 intravenously every two days (approximately 300–800 micromolar blood levels) and then underwent a 5-week follow-up for evaluation. In this study, high-dose intravenous TP5 administered three times a week enhanced immune function in patients with cutaneous and subcutaneous metastases from melanoma, with no associated side effects. Therefore, TP5 may reach relatively high concentrations at local administration sites, but concentrations in whole blood throughout the day in the average patient may be lower than those used in our tests. Relatively high doses of TP5 may be used as an immune-enhancing adjuvant to enhance the immune function of patients. Thus, although TP5 is also relatively safe at high concentrations, clinical applications to alleviate the immunosuppressive effects of morphine requires a reasonable dose and frequency of treatment to be attempted in a clinical setting.

## 4. Materials and Methods

### 4.1. Plasmids and siRNAs

The plasmid of pcDNA3.1-hMOR-WT was kindly gifted from Dr. Rui Wang. The siRNAs used in this study were purchased from Shanghai GenePharma (Shanghai, China).

### 4.2. Chemicals and Cell Culture

TP5 was purchased from ChemBest Research Laboratories Ltd. (Shanghai, China). Leukadherin-1 (LA1) and D-Tubocurarine chloride pentahydrate (TUB) were purchased from MedChemExpress (South Brunswick Township, NJ, USA). Other compounds were purchased from Sigma. THP-1 cells were cultured in RPMI-1640 medium (C22400500BT, Gibco, NY, USA) supplemented with 20% fetal bovine serum (10099-141, Gibco, NY, USA), penicillin (100 IU/mL) and streptomycin (100 μg/mL, C0222, Beyotime, Shanghai, China). A549 cells were cultured in RPMI-1640 medium (C22400500BT, Gibco, NY, USA) supplemented with 10% fetal bovine serum (10099-141, Gibco, NY, USA), penicillin (100 IU/mL) and streptomycin (100 μg/mL, C0222, Beyotime, Shanghai, China). Cells were incubated in a humidified incubator containing 5% CO_2_ at 37 °C.

### 4.3. SiRNA Mediated Knockdown

THP-1-differentiated macrophages were transfected (Lipofectamine TM RNAiMAX, 13778075, Invitrogen, CA, USA) using 100 nM of siRNA against MOR (MOR siRNA) or non-silencing siRNA (Ctrl siRNA) [52]. The medium was changed after 6 h of transfection and the culture was continued for 48 h. The OPRM knockdown efficiency of each oligomer was confirmed by RT-qPCR assay.

### 4.4. Other Plasmids Transfection

THP1-differentiated macrophages were placed in 12-well plates and transfected with 1 μg pcDNA3.1-hMOR-WT or pcDNA3.1 (Lipofectamine 3000, L3000015, Invitrogen, CA, USA) [53]. The medium was changed after 6 h of transfection and incubation was continued for 48 h. The efficiency of overexpression was confirmed by RT-qPCR assay.

### 4.5. Flow Cytometry

THP-1 cells (5 × 105 cells/well) were seeded in 12-well plates and cultured with 200 ng/mL of phorbol-12-myristate-13-acetate (PMA, HY-18739 MedChemExpress, South Brunswick Township, NJ, USA) for 48 h to differentiate into macrophages [54]. THP-1-derived macrophages were exposed to different concentrations of morphine for 48 h. The supernatant was then removed and the cells were digested with 0.25% trypsin (25200-072, Gibco, NY, USA). After washing, the cells were resuspended in 100 μL of PBS (ST476, Beyotime, Shanghai, China). For surface staining, cells were labeled with APC-CD11b (101212, Biolegend, CA, USA), PE-SIRPα (372103, Biolegend, CA, USA), and APC-CD47 (17-0479-42, eBiosciences, CA, USA). Cell suspensions were incubated with appropriate antibodies for 30 min at room temperature in the dark, followed by a washing step to remove unlabeled antibodies. Flow cytometry analysis was performed using BD LSRFortessa™ (BD Biosciences, Franklin Lakes, NJ, USA) and analyzed by FlowJo 10.6 software.

### 4.6. In Vitro Phagocytosis Assays

In vitro phagocytosis was performed using THP-1 cells, and these cells differentiated into macrophages by incubation with PMA for 48 h. THP-1 cells (2 × 105 cells/well) were seeded in 24-well plates and treated with morphine for 48 h. A549 cells were labeled with fluorescent dye probe ProtonexTM Red 600 (21207, AAT Bioquest, Pleasanton, CA, USA) in the dark for 30 min. The fluorescence of Protonex™ Red dye increases sharply as the pH decreases from neutral to acidic according to the manufacturer’s protocol [55]. Then, 2 × 105 pHrodo-red-labelled target A549 cells were added to the macrophages for 24 h at 37 °C. Co-cultured cells were collected and washed with 0.5% BSA-PBS. High-resolution images were taken on an inverted fluorescence microscope (DMI3000 B, Leica, Wetzlar, Germany) and processed in ImageJ. Phagocytic activity was analyzed by flow cytometry (BD LSRFortessa™, BD Biosciences, Franklin Lakes, NJ, USA) and analyzed by FlowJo 10.6 software.

### 4.7. RT-qPCR Assay

Total RNA was isolated and reverse transcribed using the EZ-press RNA purification kit (B0004-plus, EZBioscience, New York, NY, USA) and PrimeScript RT kit (RR047A, TaKaRa, Tokyo, Japan), respectively, according to the manufacturer’s instructions, and qPCR was performed using TB Green Premix Ex Taq (RR420A. TaKaRa, Tokyo, Japan) [56]. Relative changes in gene expression were determined using the 2-ΔΔCt method and the relative mRNA expression was normalized to GAPDH. The following primer sets were used to analyze the expression of specific genes, including CD11b, forward: 5′-ACTTGCAGTGAGAACACGTATG-3′ and reverse: 5′-TCATCCGCCGAAAGTCATGTG-3′; OPRM, forward: 5′-GCCCTTCCAGAGTGTGAATTAC-3′, and reverse: 5′-GTGCAGAGGGTGAATATGCTG-3′; IL-6, forward: 5′-ACTCACCTCTTCAGAACGAATTG-3′ and reverse: 5′-CCATCTTTGGAAGGTTCAGGTTG-3′; STAT3, forward: 5′-ACCAGCAGTATAGCCGCTTC-3′, and reverse: 5′-GCCACAATCCGGGCAATCT-3′; PDGFB, forward: 5′-CTCGATCCGCTCCTTTGATGA-3′, and reverse: 5′-CGTTGGTGCGGTCTATGAG-3′; MYC, forward: 5′-GGCTCCTGGCAAAAGGTCA-3′, and reverse: 5′-CTGCGTAGTTGTGCTGATGT-3′; ARG1, forward: 5′-TGGACAGACTAGGAATTGGCA-3′, and reverse: 5′-CCAGTCCGTCAACATCAAAACT-3′; IL1B, forward: 5′-ATGATGGCTTATTACAGTGGCAA-3′, and reverse: 5′-GTCGGAGATTCGTAGCTGGA-3′; NOS2, forward: 5′-TTCAGTATCACAACCTCAGCAAG-3′, and reverse: 5′-TGGACCTGCAAGTTAAAATCCC-3′; IFNG, forward: 5′-TCGGTAACTGACTTGAATGTCCA-3′, and reverse: 5′-TCGCTTCCCTGTTTTAGCTGC-3′; TNFA, forward: 5′-GGCGTGGAGCTGAGAGATAA-3′, and reverse: 5′-TTGATGGCAGAGAGGAGGTT-3′; GAPDH, forward: 5′-TTGGTATCGTGGAAGGACT-3′, and reverse: 5′-GGATGATGTTCTGGAGAGC-3′.

### 4.8. Cell Viability Assay

Cell viability was evaluated using the MTT assay kit (C0009, Beyotime, Shanghai, China) according to the manufacturer’s protocol [57]. A549 cells were seeded in 96-well plates at a density of 1 × 104 cells/well. THP-1 cells were seeded in 96-well plates at a density of 5 × 104 cells/well and differentiated in macrophages as described. Cells were incubated overnight at 37 °C. Next, cells were treated with morphine and incubated for 48 h. Then 10 µL of MTT was added to each well and incubated for 4 h. The absorbance was measured at 570 nm using a microplate reader (VarioskanTM LUX, Thermo, Waltham, MA, USA). Cell viability was estimated by comparing the relative absorbance values with those of the untreated samples.

### 4.9. Statistical Analysis

Data are expressed as mean ± SEM. All experiments were performed independently, at least 3 times. Statistical analyses were performed as described in each corresponding legend. Differences between two groups were assessed by unpaired two-sided Student’s *t*-test, and differences between multiple groups were assessed by one-way ANOVA and Dunnett’s post hoc test. *p* less than 0.05 was considered statistically significant. Concentration-response relationships of CD11b and MOR were obtained by measuring fluorescence intensity in response to different concentrations of morphine, and all results that were used to generate a concentration-response relationship were from the same group. The data were fit to the Hill1 equation: I/I _max_ = 1/[1 + (EC_50_/morphine) ^n^], where I is the normalized fluorescence intensity at a given concentration of ligands, I max is the maximum normalized fluorescence intensity, EC_50_ is the morphine concentration producing half of the maximum fluorescence intensity, and n is the Hill1 coefficient.

## Figures and Tables

**Figure 1 pharmaceuticals-16-00282-f001:**
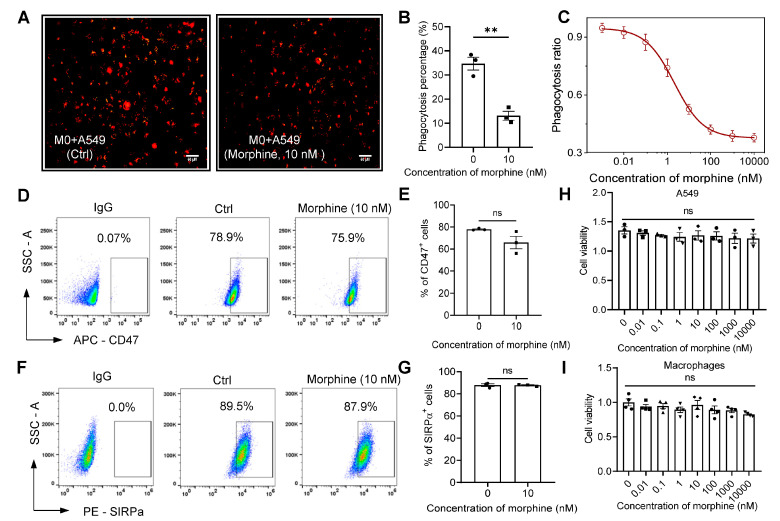
Nanomolar levels of morphine inhibit phagocytosis of A549 cells by THP-1-derived macrophages (**A**,**B**) Representative fluorescence micrographs of phagocytic activity (**A**) and statistical analysis of the mean fluorescence intensity signal in confocal images (**B**). Scale bar = 50 μm. (**C**) Flow cytometric quantification of macrophage phagocytosis under different concentrations of morphine culture (0, 0.001, 0.01, 0.1, 1, 10, 100, 1000 and 10,000 nM). (**D**–**G**) A549 cells and macrophages being labeled and analyzed for CD47- and SIRPα-expression, respectively, (**D**,**F**) by flow cytometry, and pooled data of cell surface markers of CD47 (**E**) and SIRPα (**G**). (**H**) Viability of A549 cells measured by MTT assay at different concentrations of morphine. (**I**) Macrophages viability measured by MTT assay at different concentrations of morphine (n = 3–4 independent experiments). All data are expressed as mean ± SEM.; ** *p* < 0.01 versus control, unpaired *t*-test (**B**,**E**,**G**), one-way ANOVA with Dunnett’s post-hoc test (**H**,**I**), (**H**), (F (7, 16) = 0.4882, *p* = 0.8295), (**I**), (F (7, 24) = 1.563, *p* = 0.1945. ns, not significant.

**Figure 2 pharmaceuticals-16-00282-f002:**
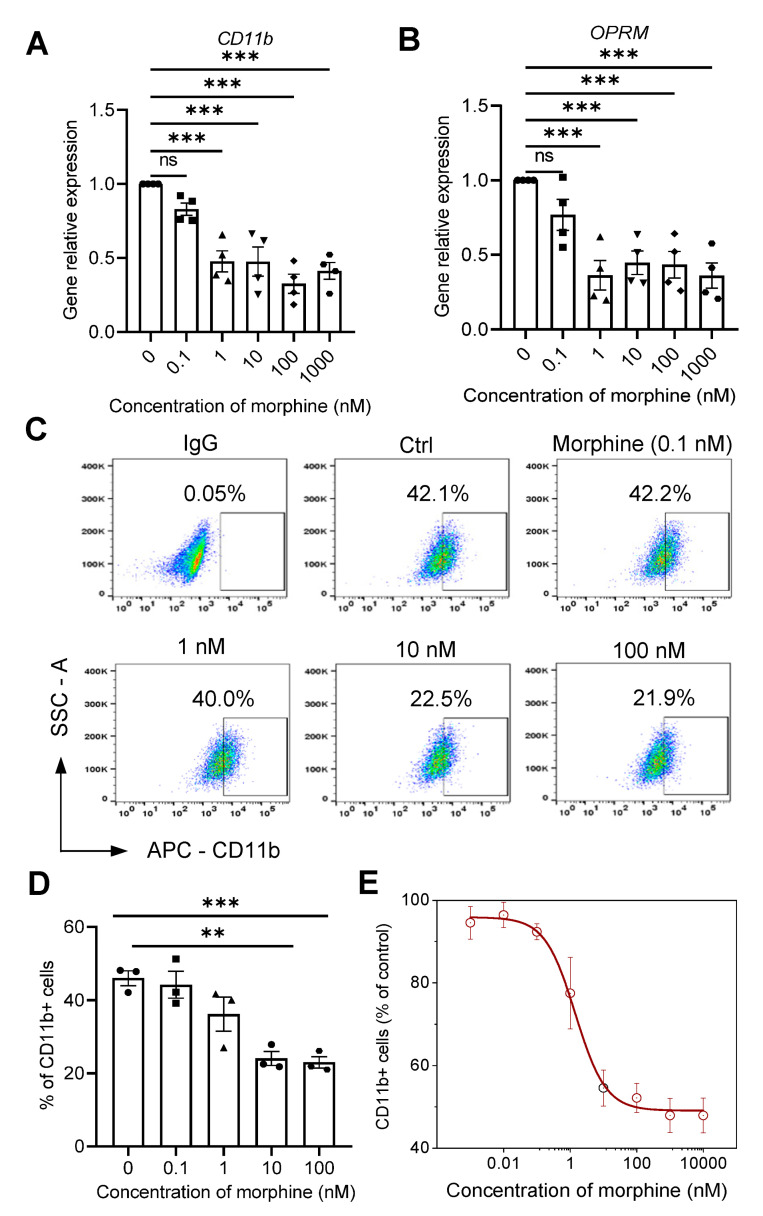
Morphine reduces the gene expression of CD11b and μ-opioid receptor (MOR) on macrophages. (**A**,**B**) RT-qPCR analysis of relative mRNA expression of CD11b (**A**) and OPRM (**B**) in macrophages. (**C**–**E**) Macrophages being labeled and analyzed for CD11b-expression (**C**) by flow cytometry, and pooled data of cell surface marker CD11b (**D**,**E**) (n = 3 independent experiments, and the solid line is fitted to *hill 1* equation). All data are expressed as mean ± SEM.; ** *p* < 0.01 and *** *p* < 0.001 versus control, one-way ANOVA with Dunnett’s post-hoc test, (**A**), (F (5, 18) = 17.83, *p* < 0.0001), (**B**), (F (5, 18) = 9.865, *p* = 0.0001), (**D**), (F (4, 10) = 13.14, *p* = 0.0005); ns, not significant.

**Figure 3 pharmaceuticals-16-00282-f003:**
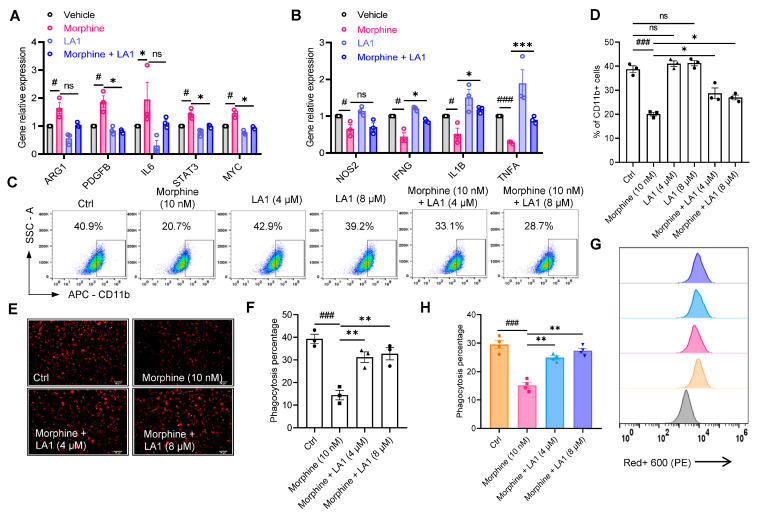
Leukadherin-1 (LA1) promotes the activation of CD11b on macrophages and alleviates the inhibitory effect of morphine on macrophage phagocytosis. (**A**,**B**) RT-qPCR analysis of relative mRNA expression of anti-inflammatory and pro-inflammatory factors ARG1, PDGFB, IL6, STAT3 and MYC (**A**), as well as NOS2, IFNG, IL1B and TNFA (**B**) in macrophages. (**C**,**D**) Flow cytometry analysis of CD11b surface expression on macrophages (**C**), and pooled data of cell surface marker CD11b (**D**). (**E**,**F**) Representative fluorescence photomicrographs of the phagocytic activity (**E**) and statistical analysis of the mean fluorescence intensity signal in confocal images (**F**). Scale bar = 50 μm. (**G**,**H**) Quantitative flow cytometry analysis of macrophage phagocytosis after incubation with different concentrations of LA1 (4 and 8 μM) and morphine (10 nM) (n = 3 independent experiments). All data are expressed as mean ± SEM.; * *p* < 0.05, ** *p* < 0.01 and *** *p* < 0.001 versus control, one-way ANOVA with Dunnett’s post-hoc test, (**D**), (F (5, 12) = 42.96, *p* < 0.05), (**F**), (F (3, 8) = 20.34, *p* = 0.0004), (**H**), (F (3, 12) = 39.07, *p* < 0.001); # *p* < 0.05 and ### *p* < 0.001 versus control; ns, not significant.

**Figure 4 pharmaceuticals-16-00282-f004:**
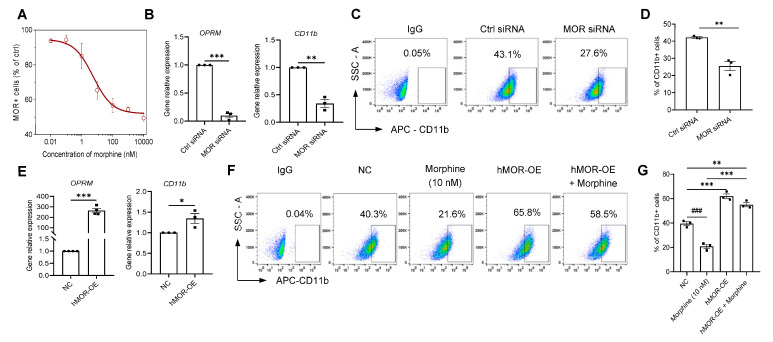
Surface expression of MOR and CD11b on macrophages is interrelated. (**A**). Flow cytometry analysis of MOR surface expression on macrophages after cells were incubated with different concentrations of morphine (the solid line is fitted to *hill 1* equation). (**B**) RT-qPCR analysis of relative mRNA expressions of CD11b and OPRM in macrophages. In transfected THP-1-derived macrophages, siRNA-mediated knockdown of OPRM also decreased the gene expression of both CD11b. (**C**,**D**) Flow cytometric analysis of siRNA-mediated alterations in CD11b expression on the surface of macrophages (**C**) and pooled data of cell surface marker CD11b (**D**). (**E**) Plasmid-mediated overexpression of OPRM in transfected THP-1-derived macrophages. (**F**,**G**) Flow cytometric analysis of the effect of morphine on CD11b surface expression in OPRM-overexpressing macrophages (**F**), and pooled data for the cell surface marker CD11b (**G**) (n = 3–4 independent experiments). All data are expressed as mean ± SEM.; * *p* < 0.05, ** *p* < 0.01, and *** *p* < 0.001 versus control, unpaired *t*-test (**B**,**D**,**E**), one-way ANOVA with Dunnett’s post-hoc test (**G**), (F (3, 8) = 110.1, *p* < 0.001); ### *p* < 0.001 versus control; ns, not significant.

**Figure 5 pharmaceuticals-16-00282-f005:**
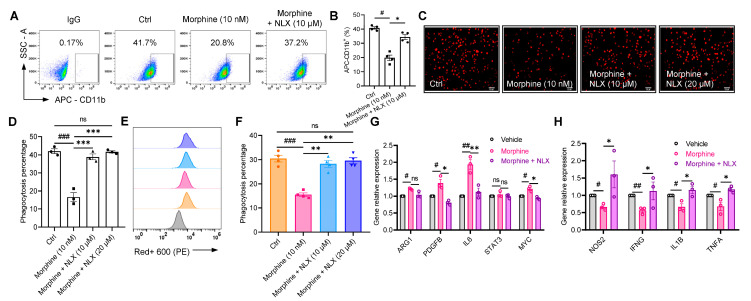
The MOR antagonist naloxone (NLX) reverses the inhibitory effect of morphine on membrane expression and function of CD11b on macrophages. (**A**,**B**) Effect (**A**) and pooled data (**B**) of NLX (10 μM) on CD11b surface expression in macrophages treated with morphine (10 nM). Macrophages were labeled and analyzed for CD11b-expression by flow cytometry. (**C**,**D**) Representative fluorescence photomicrographs of the phagocytic activity (**C**) and statistical analysis of the mean fluorescence intensity signal in confocal images (**D**). Scale bar = 50 μm. (**E**,**F**) Quantitative flow cytometric analysis of macrophage phagocytosis cultured with different concentrations of NLX (10 and 20 μM) and morphine (10 nM) (**E**), and the pooled data (**F**). (**G**,**H**) RT-qPCR analysis of relative mRNA expression of anti-inflammatory factors ARG1, PDGFB, IL6, STAT3 and MYC (**G**), and pro-inflammatory factors of NOS2, IFNG, IL1B and TNFA (**H**) in macrophages (n = 3 or 4 independent experiments). All data are expressed as mean ± SEM.; * *p* < 0.05, ** *p* < 0.01 and *** *p* < 0.001 versus control, one-way ANOVA with Dunnett’s post-hoc test, (**B**), (F (2, 9) = 47.07, *p* < 0.05), (**D**), (F (3, 8) = 54.22, *p* < 0.001), (**F**), (F (3, 12) = 31.64, *p* < 0.01); # *p* < 0.05, ## *p* < 0.01 and ### *p* < 0.001 versus control; ns, not significant.

**Figure 6 pharmaceuticals-16-00282-f006:**
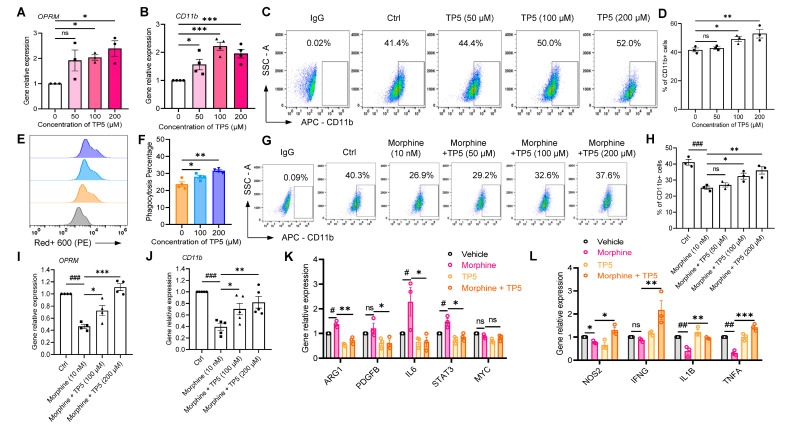
TP5 promotes the surface expression and function of CD11b on macrophages. (**A**,**B**) RT-qPCR analysis of relative mRNA expression of OPRM (**A**) and CD11b (**B**) in macrophages incubated with different treatments of TP5 (0, 50, 100 and 200 μM). (**C**,**D**) Macrophages labeled and analyzed for CD11b-expression (**C**) by flow cytometry, as well as pooled data for the cell surface marker CD11b (**D**). (**E**,**F**) Quantitative flow cytometry analysis (**E**) of macrophages phagocytosis after incubation with different treatments of TP5 (50, 100, 200 μM) and morphine (10 nM), as well as the pooled data (**F**). (**G**,**H**) Flow cytometry analysis (**G**) and pooled data (**H**) of CD11b surface expression in macrophages incubated with different treatments of TP5 (50, 100, 200 μM) and morphine (10 nM). (**I**,**J**) RT-qPCR analysis of relative mRNA expression of OPRM (**I**) and CD11b (**J**) in macrophages. (**K**,**L**) RT-qPCR analysis of relative mRNA expression of anti-inflammatory factors ARG1, PDGFB, IL6, STAT3 and MYC (**K**), and pro- inflammatory factors NOS2, IFNG, IL1B and TNFA (**L**) in macrophages (n = 3–5 independent experiments). All data are expressed as mean ± SEM.; * *p* < 0.05, ** *p* < 0.01 and *** *p* < 0.001 versus control, one-way ANOVA with Dunnett’s post-hoc test, (**A**), (F (2, 6) = 5.212, *p* = 0.0487), (**B**), (F (2, 9) = 20.07, *p* = 0.0005), (**D**), (F (3, 8) = 7.761, *p* = 0.0094), (**F**), (F (2, 9) = 14.65, *p* = 0.0015), (**H**), (F (4, 10) = 14.49, *p* = 0.0004), (**I**), (F (3, 12) = 29.04, *p* < 0.0001), (**J**), (F (3, 16) = 11.11, *p* = 0.0003); # *p* < 0.05, ## *p* < 0.01 and ### *p* < 0.001 versus control; ns, not significant.

**Figure 7 pharmaceuticals-16-00282-f007:**
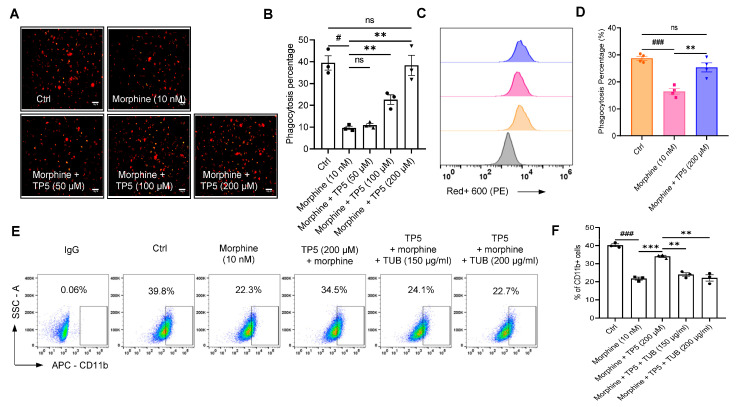
TP5 reverses the inhibitory effect of morphine on macrophage phagocytosis. (**A**,**B**) Representative fluorescence photomicrographs of the phagocytic activity (**A**) and statistical analysis of the mean fluorescence intensity signal in confocal images (**B**). Scale bar = 50 μm. (**C**,**D**) Quantitative flow cytometric analysis (**C**) of macrophage phagocytosis incubated with morphine (10 nM) and TP5 (200 μM), as well as pooled data (**D**). (**E**,**F**) Flow cytometry analysis (**E**) of CD11b surface expression and pooled date (**F**) on macrophages treated with morphine (10 nM) and TP5 (200 μM) or in combination with the AchR inhibitor D-Tubocurarine chloride pentahydrate (TUB, 150 and 200 μg/mL) (n = 3–4 independent experiments). All data are expressed as mean ± SEM.; ** *p* < 0.01 and *** *p* < 0.001 vs. control, one-way ANOVA with Dunnett’s post-hoc test, (**B**), (F (4, 10) = 27.15, *p* < 0.0001), (**D**), (F (2, 9) = 27.44, *p* = 0.0001), (**F**), (F (4, 10) = 61.76, *p* < 0.01); # *p* < 0.05 and ### *p* < 0.001 versus control; ns, not significant.

## Data Availability

Data is contained within the article.

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
