# Peer review of "Altered Membrane Expression and Function of CD11b Play a Role in the Immunosuppressive Effects of Morphine on Macrophages at the Nanomolar Level"

_pharmaceuticals, 2023, doi:10.3390/ph16020282_

Round 1
Reviewer 1 Report
The manuscript by Yu P.C. et al, “Altered membrane expression and function of CD11b play a role in the immunosuppressive effects of morphine on macrophages at the nanomolar level” reports, at the cell level, 0.1-10 nM of morphine attenuated macrophage phagocytosis by inhibiting CD11b expressing and function on macrophages via mu-opioid receptor. In addition, thymopentin (TP5) comnined with morphine, could reverse the immunosuppression of morphine on macrophages. And the present results are interesting and important for the readers in the field. However, it remains several important points to revise.
MAJOR ISSUES:
1- The authors state that the inhibitory effect of morphine on macrophage phagocytosis only basing on the celluar data. However, the in vivo blood concentration of morphine in the clinical was used in the Discussion. It should be better to add in vivo experiments by using rodents to prove the hypothesis.
2- It should be better to discuss the current results. For example, what is the possibel mechanisms related to the functional interaction between TP5 and morphine?
MAJOR ISSUES:
1- Please check the information of AUTHORS, “Ye Yu and *?”
2- Morphine is rarely used orally in the clinic, but is often intravenous or subcutaneous administration as an analgesic. Its blood concentrations after intravenous or subcutaneous administration should be calculated.
Author Response
We thank the reviewer for his/her concise summaries and supportive comments on our work, and we have made changes to clarify the issues raised by the reviewer. Below is our detailed point-by-point response to the reviewer' comments.

Reviewer 2 Report
Reviewers’ comments:
Morphine was widely used in clinical analgesia, while clinical use of morphine may be accompanied by increased immunosuppression, vulnerability to infection and postoperative tumor metastatic recurrence. In this manuscript, Peng-Cheng Yu et al. found that morphine at nanomolar level inhibited CD11b expression and function on macrophages via a μ-opioid receptor (MOR)-dependent mechanism. By using antagonist of MOR receptor and knocking down or overexpressing MOR on macrophages, they found that morphine can regulate macrophage phagocytosis by inhibiting the surface expression and function of macrophage CD11b through the membrane expression and activation of MOR. Besides, TP5, as an immune-enhancing adjuvant in clinical practice, when combined with morphine, reversed the inhibition of macrophage phagocytosis by morphine through mechanisms that promote membrane expression of CD11b and modulate its downstream signaling. The evidence, in terms of data quality and the comprehensive approach used, supports the authors' conclusions well. Not only do they propose a new mechanism for the action of CD11b in the immunosuppressive process induced by morphine, but they also provide a reversal of this process by TP5 used clinically and therefore is recommended for acceptance for publication, but the following points remain to be improved.
Minor concerns:
1、 TP5 combined with morphine can reversed the immunosuppression caused by morphine, while whether the analgesic effects of morphine will be affected by TP5?
2、 The authors propose that TP5, when used in combination with morphine, reverses the inhibition of macrophage phagocytosis by morphine through a mechanism that promotes membrane expression of CD11b and regulates its downstream signals (e.g., NOS2, IFNG, IL1B, and TNFA, as well as AGR1, PDGFB, IL6, STAT3, and MYC). In the case of nAchR receptors mediating this process, the involvement of ion channels in this process is very interesting, but this manuscript does not provide more information on how this mechanism is understood, so it is suggested that the authors discuss possible mechanisms at least in the introduction or discussion section.
3、 The concentration of morphine acting on macrophage CD11b is very low compared to the same concentration of therapeutic dose in clinical application, but the concentration of TP5 is higher compared to clinical treatment. This is also described by the authors in the manuscript, and it would be better to elaborate further.
4、 There are some details that need to be refined.
For example,
Fig. 4C, siRNACTRL, siRNAMOR, Fig. 4F, MOR-OE, OPRM.
Figure 7E, TUB, etc., are easily misunderstood or mixed up in a meaningless way, and it is suggested to adjust them and address the meaning of the abbreviations in the figure legends.
Also "5-10 mg/dose" ". I/I max = 1/ [1 + (EC50/morphine)]", etc.
Author Response

(The authors gave the same response as above.)

Reviewer 3 Report
In this manuscript, Yu et.al reported that morphine acts on the mu-opioid receptors in the macrophages regulating the expression and function of CD11b, which in turn induced immunosuppression by reducing macrophage phagocytosis. This is an interesting study. Opioid analgesics are the major analgesics in late-stage cancer patients for the treatment of cancer-induced pain or chemotherapy-induced pain. The immunosuppression of morphine may increase the risk of infection and negatively affect cancer treatment. This study provides new mechanistic insight into the immunosuppression effect of opioid analgesics. The manuscript is well-written, and the data is well-organized, supporting their conclusions.
I have only one question about the over-expression and siRNA-mediated knockdown of the OPRM experiments (Figure 2). Figure 2D shows that the knockdown of the MOR, CD11b also has a significant decrease. The knockdown experiment should be similar to the antagonist, knockdown may block the effects of morphine. But the author found knockdown MOR itself decreased CD11b expression. What is the potential mechanism? In Figure 2G, when MOR was over-expressed, the CD11b was also increased, and morphine seems cannot inhibit the expression of CD11b in MOR over-expression macrophages. Theoretically, the MOR over-expression makes morphine have more receptors on macrophages, it may enhance the inhibition of CD11b. These results indicate that the expression of MOR itself can directly regulate the expression of CD11b, which may be independent of its activation by its agonist in these over-expression and siRNA-mediated knockdown experiments. These are interesting findings but need more discussion.
Author Response

(The authors gave the same response as above.)

Reviewer 4 Report
The topic of this article is exciting, and I really appreciate your paper. For more than a decade a lot of publications warned about the potential risk of opioids for cancer patients by increasing the tumor growth.Your research is fundamental and demonstrate the mechanism of immunosuppression Inhibiting phagocytosis by macrophages. you also clearly demonstrate the role of CD11b via Mor receptors. Finally, you also demonstrate that Naloxone and Thymopentin can reverse the inhibitory effect of morphine .The methodology is accurate and well describedThe outcomes are clearly presented and figures absolutely illustrate your results .
Moreover , I also appreciated your comment ( L459 ) on the potential relationship of opioid use for non-cancer pain .Finally, only TP5 could be use in clinical practice to reverse this immunosuppressive effect of morphine as Naloxone should reverse analgesic effect.
Really Great job
Author Response
We really appreciate the positive comments and compliments from the reviewer, and we will continue to study this issue extensively.
